# Systematic Investigation of the Synergistic and Antagonistic Effects on the Removal of Pyrene and Copper onto Mesoporous Silica from Aqueous Solutions

**DOI:** 10.3390/ma12030546

**Published:** 2019-02-12

**Authors:** Haiyan Li, Mengyun Zhai, Hongrui Chen, Chaohong Tan, Xiaoran Zhang, Ziyang Zhang

**Affiliations:** 1Beijing Engineering Research Center of Sustainable Urban Sewage System Construction and Risk Control, Beijing University of Civil Engineering and Architecture, Beijing 100044, China; lihaiyan@bucea.edu.cn (H.L.); mengyunzhai@126.com (M.Z.); chaohongtan@126.com (C.T.); 2CRRC Environmental Science & Technology Cooperation, Beijing 100067, China; hr630@126.com; 3Key Laboratory of Urban Stormwater System and Water Environment, Ministry of Education, Beijing University of Civil Engineering and Architecture, Beijing 100044, China; zhangxiaoran@bucea.edu.cn

**Keywords:** synergistic effects, antagonistic effects, mesoporous silica, pyrene, copper

## Abstract

Polycyclic aromatic hydrocarbons (PAHs) and heavy metals have attracted greater attention due to their single or complex risks. It is urgent to find useful methods to remove these two pollutants together. In this study, SBA15 and MCM-41 were selected and used for the simultaneous removal of pyrene and copper from aqueous solution. Batch experiments were conducted systematically by investigating the adsorption behavior and effects including kinetics, isotherms, ionic strength and pH effects. Experimental results showed that the Langmuir and pseudo-second-order model fitted the adsorption behavior better. The solution pH values and ionic strength affected the adsorption behavior greatly. Furthermore, the synergistic or antagonistic effects could be observed on the adsorption of pyrene and copper onto MCM-41 and SBA15, respectively. The synergistic and antagonistic effects of pyrene and copper onto mesoporous silica may be attributed to the size of pyrene–copper complex and the average pore size of adsorbents. With the higher pore size, the complex would be adsorbed onto the inner surface of MCM-41 which showed synergistic effect on the adsorption of pyrene and copper. This study shows new guidelines and insight into the study of adsorption behavior of PAHs and heavy metals from aquatic environments.

## 1. Introduction

As two typical unconventional pollutants, polycyclic aromatic hydrocarbons (PAHs) and heavy metals have attracted increasing concern in recent years. PAHs are persistent organic compounds which are carcinogenic, teratogenic, mutagenic and have a low bioavailable fraction [1]. As one of the typical PAHs, pyrene was selected as a priority pollutant by the US Environmental Protection Agency. Furthermore, pyrene has been widely used as an indicator compound for the overall burden of all PAHs [2]. As a persistent material, heavy metals could not be eliminated by microbial decomposition. Heavy metals could be enriched by the food chain and may cause significant pathological changes to organisms [3]. PAHs and heavy metals could enter the aquatic environment from various sources. Results showed that PAHs and heavy metals could be detected in various aquatic environments such as rivers, lakes and oceans [4]. Although the concentrations of both PAHs and heavy metals were low (µg/L to ng/L), the toxicity of PAHs and heavy metals could not be neglected. Furthermore, results showed that the PAHs and heavy metals could form complex compounds by cation-π bonding, and that the toxicity of formed complex was larger than their individual toxicity which enhance the co-toxicity of PAHs and heavy metals [5]. However, fewer studies have dealt with the removal of PAHs and heavy metals together from aqueous solutions. Consequently, it is important to find effective ways to eliminate PAHs and heavy metals from water environments together.

Previous studies have shown that PAHs and heavy metals are difficult to eliminate by coagulation, flocculation, sedimentation, filtration and other traditional physicochemical methods. In recent years, many approaches have been used to eliminate PAHs and heavy metals; for instance adsorption, photocatalysis oxidation, biodegradiation, advanced oxidation process and extraction techniques. Extraction techniques and advanced oxidation processes usually need high investment and high maintenance costs, while biodegradation and photocatalysis oxidation processes may release some secondary by-products which are carcinogenic and mutagenic compounds. Among these methods, adsorption is attractive due to the multiple benefit of high removal capacity, easy operation, energy savings, quick and wide application [6]. A large amount of adsorbents was used to eliminate PAHs and heavy metals from aqueous solutions, for example activated carbon [7], zeolites [8], carbon nanoporous materials [9], porous glass [10], biosorbents [6], chitosan [11], montmorillonite [12], slag [13] and mesoporous materials [14]. Although many of them had good effects on PAHs and heavy metals, fewer reports focused on the synergistic or antagonistic effects on the elimination of these two compounds. There has been no mention of investigating the mechanism to eliminate the complex compounds formatted by PAHs and heavy metals. Hence, the investigation of synergistic or antagonistic effects on the adsorption of PAHs and heavy metals has been attracting attention.

As a typical mesoporous adsorbent, mesoporous silica has a high potential application in the removal of pollutants due to the unique large surface area, large number of adsorption sites, well-defined pore sizes and pore shape [15,16]. Moreover, it has a cylindrical structure and symmetry pores make it better-suited to application in separations and adsorption [14,17]. A large number of mesoporous materials have already been used as efficient adsorbents for organic and inorganic pollutants from aqueous solutions. Hu et al. [18] found that MCM-41 had a great adsorption capacity of phenanthrene and could be used for the removal of phenanthrene from aqueous solution. Batati et al. [10] used NH_2_-SBA15 organic–inorganic nanohybrid materials to eliminate PAHs from wastewater and found that these materials also show high adsorption capacity after several cycles of adsorption–desorption. Mureseanu et al. [19] used N-propylsalicylaldimino-functionalized SBA15 to eliminate heavy metals and found the adsorbent has remarkable adsorption capacities in the removal of copper, cobalt, nickel and zinc. However, most of these results were focused on the single system and not investigated the synergistic or antagonistic effects on the removal efficiency, which inhibit the application of mesoporous silica in the elimination of PAHs and heavy metals from aqueous solutions.

In this study, two typical mesoporous silica (SBA15 and MCM-41) were selected to investigate the adsorption of PAHs and heavy metals from aqueous solution together. Pyrene and copper were selected as the pollutants due to the large amount detection rate in environment. The surface morphology, structure, specific surface area, pore volume and size distributions of SBA15 and MCM-41 were investigated. The adsorption of pyrene and copper onto SBA15 and MCM-41 were investigated by single and binary systems. To investigate the adsorption behavior and effects such as kinetics, isotherms, ionic strength and pH effect, batch experiments were conducted in single and binary systems. This research provides new ideas for the elimination of PAHs and heavy metals in aquatic environments simultaneously.

## 2. Materials and Methods

### 2.1. Materials and Chemicals

Pyrene (99%) was purchased from the Sigma Aldrich Chemical Co. (Milwaukee, WI, USA). Tetraethyl silicate (TEOS, 98%), PluronicP123 (PEO20 PPO70 PEO20, Mav = 5800) and cetyl trimethyl ammonium bromide (CTAB, 99%) were purchased from Alfa Aesar Corp. (Heysham, UK). UPLC grade of Acetonitrile and methanol were purchased from Fisher Scientific Corp. (Shanghai, China). Other chemicals including HCl, NaOH, CuCl_2_·2H_2_O, KCl and ethanol were all of analytical grade. All solutions were prepared in high-purity water (Milli-Q).

### 2.2. The Preparation of Adsorbents

SBA15 was synthesized by hydrothermal method with Tetraethyl silicate (TEOS) as the silicon source and P123 as template [20]. The synthesis method showed as follows: firstly, dissolved 2 g P123 into 70 mL HCl solution (2 mol L^−1^) at 35 °C. Then added 4.5 g TEOS into solution and continuous stirring for 20 h at 35 °C. Next transferred the mixture into autoclave for 24 h at 100 °C. Subsequently, recovered resultant through filtered, washed and dried at 25 °C for 12 h. To remove the organic component, calcination was conducted in muffle at 540 °C, the SBA15 mesoporous material was finally obtained.

MCM-41 was prepared by a direct-synthesis approach under alkaline condition using Tetraethyl silicate (TEOS) as the silicon source and cetyl trimethyl ammonium bromide (CTAB) as the template [21]. The synthesis method showed as follows: firstly, 0.84 g CTAB was added into deionized water. Then adjusted the solution pH value about 11.5 with 1.0 M NaOH. After stirring for 15 min, added 3.605 g TEOS to the solution under vigorous stirring and continuous for 1 h. After stirring continuously for 20 h at 25 °C, the recovered resultant was filtered, washed and dried at 100 °C for 12 h. Finally, calcined the resultant in muffle at 540 °C for 10 h to remove CTAB, further obtained the MCM-41 mesoporous material.

### 2.3. Characterization Techniques

The surface morphologies of the samples were examined by scanning electron microscopy (SEM, FEI Quanta 200, Eindhoven, Netherlands). The phases and micrographs of mesoporous materials were analyzed by transmission electron microscope (TEM, JEOLJEM-2000cx, Beijing, China). X-ray powder diffraction (XRD) patterns were obtained by Bruker D8 (Karlsruhe, Germany) with Cu-Kα radiation. The data were collected with 2θ = 0.2–5° with a scan step of 0.5° min^−1^. N_2_ adsorption-desorption isotherms of SBA15 and MCM-41 samples were collected at 77 K on ASAP 2020 HD88 (Micromeritics). The specific surface area was calculated by Brunauer-Emmett-Teller (BET) method [22], pore volume and pore size distribution were calculated by BJH method [23].

### 2.4. Adsorption Experiments

The adsorption behaviors of pyrene and copper onto SBA15 and MCM-41 in single and binary systems were conducted by batch experiments including adsorption kinetics, isotherms, pH and ionic strength effect. The experiments were carried out at 25 °C with an ionic strength of 0.01 mol L^−1^ KCl. All the experiments were performed in duplicate and the values listed were calculated by the average and standard deviation. The standard deviations were calculated by Equation (1).
(1)$$S=\sqrt{\frac{1}{N}\sum_{i=1}^{N}(\bar{x}-x_{i}}{)}^{2}$$
where *S* is the standard deviation, $\bar{x}$ is the average of each sample, *x_i_* is the *i*th sample. 

In the kinetics experiments, three sets of solutions of 500 mL were prepared with a pyrene initial concentration of 0.1 mg L^−1^, copper initial concentration of 20 mg L^−1^ and simultaneous pyrene initial concentration of 0.1 mg L^−1^ and copper initial concentration of 20 mg L^−1^. Then, 0.25 g SBA15 and MCM-41 were added into the solution and the mixtures were shaken with a speed of 170 rpm. At 0, 5, 10, 15, 25, 35, 45, 60, 75, 90, 120, 150, 180, 210, 270, 330, 390, 510, 630 min, the samples were taken and centrifuged for 2 min at 5000 rpm. The concentrations of the pyrene and copper in the samples were analyzed and the equilibrium adsorption amount of pyrene and copper onto the SBA15 and MCM-41 were calculated by Equation (2).
(2)$$Q_{e}=V(C_{0}-C_{e})/m$$
where *C*_0_ is the initial concentration while *C_e_* is the equilibrium concentration of pyrene and copper (mg L^−1^), *Q_e_* (mg g^−1^) is the equilibrium adsorption capacity of the adsorbents, *V* is the volume of pyrene and copper solution (L), *m* is the weight of the adsorbent (g).

The adsorption isotherms of pyrene and copper onto SBA15 and MCM-41 in single and binary systems were conducted. For the adsorption of pyrene, the 10 mg L^−1^ pyrene (methanol as a solvent) were prepared. Then, the stock solution of pyrene were diluted to 0.025, 0.050, 0.075, 0.100, 0.125, 0.150 mg L^−1^ which contains 0 or 20 mg L^−1^ copper. For the adsorption of copper, the solution of copper with the concentration of 5, 10, 15, 20, 25, 30, 35, 40, 50 mg L^−1^ which contains 0 or 0.1 mg L^−1^ pyrene. Next, 25 mL mixed solution was added into glass vials and the adsorbents were added with a dosage of 0.5 g L^−1^. Finally, the mixture solutions were shaken with 170 rpm at 25 °C for 24 h until the equilibrium achieved. We centrifuged the solutions and took the supernatant to analyze the concentrations of pyrene and copper. 

The effect of pH and ionic strength on the adsorption of pyrene and copper onto SBA15 and MCM-41 were investigated. We prepared 25 mL solutions for three sets as kinetics experiments. Added SBA15 and MCM-41 into the solutions with the dosage of 0.5 g L^−1^. The solution pH was kept at desired values in the range of 1.0–5.0 using HCl and NaOH solution during the adsorption process. The solution ionic strength was kept at desired values in the range of 0–0.1 mol L^−1^ using KCl during the adsorption process. Other experimental conditions were the same as the isotherm experiments.

The amount of 0.025-g prepared adsorbents (SBA15 and MCM41) were added into a 25 mL solution. The solution pH was kept at desired values in the range of 2.0–13.0 using HCl and NaOH. The mixtures were shaken for 24 h to reach the equilibrium and the Zeta potential was measured using a Nano Particle Sizing & Zeta potential Analyzer (Beckman Coulter, U.S.A).

The concentration of pyrene in the solution were analyzed by Ultra Performance Liquid Chromatography (UPLC: Acquity, Waters, Reverse-phase column C18, 100 mm × 2 mm × 1.5 μm) at 237 nm. Isocratic elution was performed at a flow rate of 0.4 mL min^−1^ using the methanol/0.01 CH_3_COOH (90:10) mobile phase. The concentration of copper in the solution were determined by Atomic Absorption Spectrophotometry (Hitachi Z-2010, Tokyo, Japan)

## 3. Results and Discussion

### 3.1. Structure, Textural and Morphological Properties of Adsorbents

Representative TEM and SEM micrographs of SBA15 and MCM-41 are shown in Figure 1. As depicted in Figure 1a, SBA15 shows uniform pore size distribution and ordered three-dimensional hexagonal channels, which is typical of a mesoporous material [15]. Furthermore, Figure 1c shows that SBA15 consists of many rope-like domains with a uniform length of 1 μm. These results suggest that the SBA15 was successfully prepared. As shown in Figure 1b, we presented the agglomerated spherical particles that are a typical feature of MCM-41 materials [22]. Figure 1d shows that MCM-41 has a uniform pore size distribution and ordered hexagonal channels, which means that the MCM-41 was successfully prepared.

Figure 2 shows the small-angle X-ray diffraction patterns for SBA15 and MCM-41. SBA15 and MCM-41 both have two characteristic peaks associated with (100) and (200) diffraction planes. The sharpness of characteristic peaks suggest that the degree of regularity of the pores in silicon mesoporous. SBA15 exhibited a well-defined diffraction peak at 0.9° and 1.6°, which could be indexed to the (100) and (110) planes of the porous structure and associated with two-dimensional hexagonal P6 mm symmetry, suggesting that SBA15 is well-synthesized [24]. MCM-41 has a well-defined diffraction peak (100) at 2.0°, which is the characteristic diffraction peak of hexagonal mesoporous materials. In addition, MCM-41 exhibit legible diffraction peaks in the range of 3–5°, which could be indexed to the (110) planes, suggesting MCM-41 have long distance hexagonally arranged channels [25]. Compared with MCM-41, SBA15 have even higher (100) reflection intensity, implying some shape sorts in the hexagonal arranged channels.

The N_2_ adsorption/desorption isotherms and corresponding BJH pore size distributions are depicted in Figure 3. As shown in Figure 3a, according to the IUPAC classification, N2 adsorption/desorption isotherms of SBA15 and MCM-41 are classical type IV adsorption isotherms with H1 hysteresis loops which indicated that the prepared materials had a well-defined pore structure. Furthermore, as shown in Figure 3b, the adsorbents have uniform narrow pore size distribution which the average pore size are 4.12 nm and 3.74 nm for SBA15 and MCM-41, respectively. The pore structural properties (BET surface areas, pore volumes and sizes) are showed in Table 1. These data show that MCM-41 had smaller pore size but larger BET surface area and pore volume than the SBA15.

### 3.2. The Adsorption Kinetics of Pyrene and Copper onto SBA15 and MCM-41

Figure 4 describe the adsorption kinetics of pyrene and copper onto SBA15 and MCM-41. For both SBA15 and MCM-41, the adsorption of pyrene and copper are fast first then a slow process and eventually reached equilibrium in both single and binary systems. The equilibrium time of pyrene and copper are about 100 and 150 min for MCM-41. The equilibrium time of pyrene and copper onto SBA15 was about 200 and 350 min. The rapid adsorption rate of pyrene and copper onto SBA15 and MCM-41 could be attributed to the large number of useful adsorption sites on the surface and pore structure of mesoporous molecular sieves. Furthermore, the synergistic and antagonistic effects of pyrene and copper are observed for MCM-41 and SBA15, respectively.

In order to study the adsorption kinetics further, pseudo-first-order and pseudo-second-order models were used to predict the adsorption mechanism, the equations were expressed by Equations (3) and (4), respectively [26].
(3)$$\ln\left(Q_{e}-Q_{t}\right)=\ln Q_{e}-k_{1}t$$
(4)$$t/Q_{t}=1/k_{2}Q_{e}^{2}+t/Q_{t}$$
where *Q_t_* (mg g^−1^) and *Q_e_* is the adsorption capacity onto SBA15 and MCM-41 at time *t* and equilibrium, *k*_1_ (min^−1^) is the kinetics rate constants of the pseudo-first-order model and *k*_2_ (mg g^−1^ min^−1^) is the kinetics rate constants of pseudo-second-order model, respectively.

As shown in Table 2, the correlation coefficients (*R*^2^) for pseudo-second-order model are relatively higher than that of the pseudo-first-order model, which suggested that the adsorption process of pyrene and copper onto SBA15 and MCM-41 may be chemisorption. The adsorption of pyrene and copper onto mesoporous silica were probably occurred by surface complexation reactions between copper, pyrene and the sorption sites on the adsorbents [26].

### 3.3. The Adsorption Isotherms of Pyrene and Copper onto SBA15 and MCM-41

Adsorption isotherm respond to the adsorption capacity of adsorbents at different equilibrium concentrations. Adsorption isotherms of pyrene and copper in single and binary systems on SBA15 and MCM-41 are exhibited in Figure 5. Obviously, the adsorbing capacity of pyrene and copper rise as the equilibrium concentration increases. All adsorption isotherms showed a sharp initial slope. These were because the concentration of contaminants were low, while the available adsorption sites in adsorbents were large. As the equilibrium concentrations of pollutants increased further, the adsorption sites were gradually occupied and the capacity of adsorbents increased gradually as well. Furthermore, the antagonistic and synergistic effects on the removal of pyrene and copper onto SBA15 and MCM-41 could be observed. The presence of copper decreased the adsorption capacity of pyrene onto SBA15 (Figure 5a) and increased the adsorption capacity of pyrene onto MCM-41 (Figure 5b), which was consistent with the adsorption of copper on SBA15 and MCM-41. These results are consistent with the adsorption kinetics as mentioned before.

To investigate the adsorption process further, the Langmuir and Freundlich model equations were used to describe the adsorption isotherm in detail which were commonly represented as Equations (5) and (6), respectively [27]:
(5)$$Q_{e}=Q_{\max}bC_{e}/(1+bC_{e})$$
(6)$$Q_{e}=K_{f}{C_{e}}^{1/n}$$
where *Q*_max_ is the maximum adsorption capacity (mg g^−1^), *b* is an equilibrium constant associated with the adsorption energy, *K_f_* is the Freundlich adsorption constant and *n* is related to the degree of deviation from liner adsorption.

As shown in Table 3, the Langmuir model fitted the adsorption process better (*R*^2^ = 0.92~0.99) than the Freundlich model in both single and binary systems, which may indicate the adsorption of pyrene and copper on mesoporous silica were monolayer molecular adsorption. Further, the values of 1/*n* were hardly larger than 1.0, suggesting that the adsorption of pyrene and copper onto SBA15 and MCM-41 were favorable. The *Q*_max_ calculated by Langmuir model indicating that the synergistic and antagonistic effects of pyrene and copper were observed for MCM-41 and SBA15, respectively. For SBA15, the *Q*_max_ of copper decreased from 10.28 mg L^−1^ to 3.08 mg L^−1^ in the absence or presence of pyrene. While for MCM-41, the *Q*_max_ of copper increased from 10.53 mg L^−1^ to 15.09 mg L^−1^ with and without pyrene. The same phenomenon was also observed for the adsorption of pyrene onto adsorbents with and without copper. These results show that MCM-41 can adsorb pyrene and copper simultaneously and effectively in solution.

### 3.4. Effects of pH

Figure 6 shows the influences of solution pH on the adsorption of pyrene and copper onto SBA15 and MCM-41 in single and binary systems. As shown in Figure 6a,b, with the solution pH values increased from 2.0 to 5.0, the adsorption capacity of pyrene onto SBA15 and MCM-41 decreased from 186.31 µg g^−1^ and 64.84 µg g^−1^ to 81.35 µg g^−1^ and 40.18 µg g^−1^ respectively in single systems. On the contrary, as shown in Figure 6c,d, with the increase of pH values, the adsorption capacity of copper increased from 0.67 mg g^−1^ and 3.29 mg g^−1^ to 4.04 mg g^−1^ and 9.61 mg g^−1^ in single systems. The same results were observed in binary systems. These results could be explained by the electrostatic interaction between adsorbents and contaminants. As known, the change of pH values affects the electrical conductivity of the surface of SBA15 and MCM-41 greatly. In order to investigate the effect of pH values on the surface charge, the zeta potential of SBA15 and MCM-41 were conducted and the results are listed in Figure 7. When the pH values increased from 2.0 to 5.0, the negative charges on the surfaces of SBA15 and MCM-41 increased. As a nonionic compound, pyrene has an electron-rich π ring (as a donor of electrons). Therefore, the electrostatic attraction between pyrene and the two adsorbents is weakened, resulting in a decrease in the adsorption capacity for pyrene [10]. However, copper showed positive charged in the pH values of 2.0–5.0. Due to the increasing electrostatic attraction between copper and adsorbents, the adsorption capacity of copper increased [16,19].

Furthermore, as shown in Figure 6a,c, compared with the single systems, the adsorption capacity of pyrene and copper onto SBA15 decreased in binary systems in all solution pH values, which suggested the antagonistic effects on the adsorption of pyrene and copper. As shown in Figure 6b,d, the adsorption capacity of pyrene and copper onto MCM-41 increased in binary systems in all solution pH values. In other words, MCM-41 can synergistic remove pyrene and copper from aqueous solutions.

### 3.5. Effects of Ionic Strength

Figure 8 shows the effect of ionic strength on the adsorption of pyrene and copper onto SBA15 and MCM-41. As shown in Figure 8a,b, the adsorption capacity of pyrene onto SBA15 and MCM-41 increased from 38.44 µg g^−1^ and 18.58 µg g^−1^ to 115.51 µg g^−1^ and 83.31 µg g^−1^, respectively, with the KCl concentration increasing from 0 to 0.1 mol L^−1^. These results could be attributed to the salting-out effect. As an nonionic compound with a strong hydrophobic effect, the adsorption mechanism of pyrene was mainly caused by the hydrophobic interaction [28]. When the ionic strength increased from 0 to 0.1 mol L^−1^, the solubility of pyrene and the activity coefficients in solutions decreased, which induced the adsorption of pyrene onto SBA15 and MCM-41. In contrast, as shown in Figure 8c,d, the adsorption capacity of copper onto SBA15 and MCM-41 decreased from 11.12 mg g^−1^ and 16.29 mg g^−1^ to 1.35 mg g^−1^ and 4.29 mg g^−1^ when the KCl concentration increased from 0 to 0.1 mol L^−1^. The observed results could be attributed to the chelation between copper and Cl^−^. Furthermore, with the ionic strength increasing, Cl^−^ could react with copper to form a stable complex [29]. The same results were observed in binary systems. Furthermore, as shown in Figure 8a,c, compared with the single systems, the adsorption capacity of pyrene and copper onto SBA15 decreased in binary systems in all ionic strengths, which confirmed the antagonistic effects of pyrene and copper. As shown in Figure 8b,d, the adsorption capacity of pyrene and copper onto MCM-41 increased in binary systems in all ionic strengths, which suggested the synergistic effect of pyrene and copper. These results were consistent with the conclusion mentioned before.

### 3.6. Adsorption Mechamism

As we know, both SBA15 and MCM-41 were silicon-based mesoporous adsorbents; the main component was almost the same. However, as other researcher confirmed, except for mesoporous, SBA15 also contained some micropores [30]. Kosuge et al. [31] found that in the adsorption of VOCs onto mesoporous silica, the adsorption process was preferable to occur through micropores on SBA15 and through mesoporous on MCM-41. In this study, especially in binary systems, due to the complexation between pyrene and copper, there are three kinds of adsorbates in the solutions, such as copper, pyrene and their complex compounds. As shown in Figure 9, the average pollutant size followed the order of complex compounds > pyrene > copper. After the addition of mesoporous silica adsorbent, the adsorbates are adsorbed firstly onto the outer surface of the sorbent, then transform into the inner surface of the adsorbent. For SBA15, due to the large size of the composite contaminants, the pore of the outer surface may be blocked by complex compounds which prevent the transformation of pollutants into the inner surface of SBA15, and induced the decreasing of the adsorption capacity of pyrene and copper. In other words, the antagonistic effects were observed. For MCM-41, due to the larger pore structure, the complex compounds of pyrene and copper were easily transformed into the inner surface of adsorbents, which enhance the adsorption capacity of pyrene and copper, and the synergistic effects were observed. The adsorption schematic diagram of copper and pyrene onto SBA15 and MCM-41 are shown in Figure 9.

## 4. Conclusions

In this study, two typical mesoporous silica SBA15 and MCM-41 were prepared and used to investigate the simultaneous removal of pyrene and copper from aqueous solution. Results showed that the adsorption of pyrene and copper onto mesoporous silica was fast in both single and binary systems. The Langmuir model fitted the adsorption isotherms better than the Freundlich model, which suggested that the adsorption was probably a monolayer molecular adsorption. The kinetics of pyrene and copper onto SBA-15 and MCM-41 could be satisfactorily described by a pseudo-second-order model which suggested that the adsorption may be chemisorption. The solution pH values and ionic strength greatly influence the adsorption process. Furthermore, the synergistic effects could be observed on the adsorption of pyrene and copper onto MCM-41, while the antagonistic effects could be observed on the adsorption of pyrene and copper onto SBA15. The different pore sizes have a great influence on the adsorption process of pyrene, copper and their composite contaminants. This study can provide new insights into the removal of PAHs and heavy metals from aqueous solutions. This study could provide new ideas for the synergistic removal of PAHs and heavy metals in water environments. 

## Figures and Tables

**Figure 1 materials-12-00546-f001:**
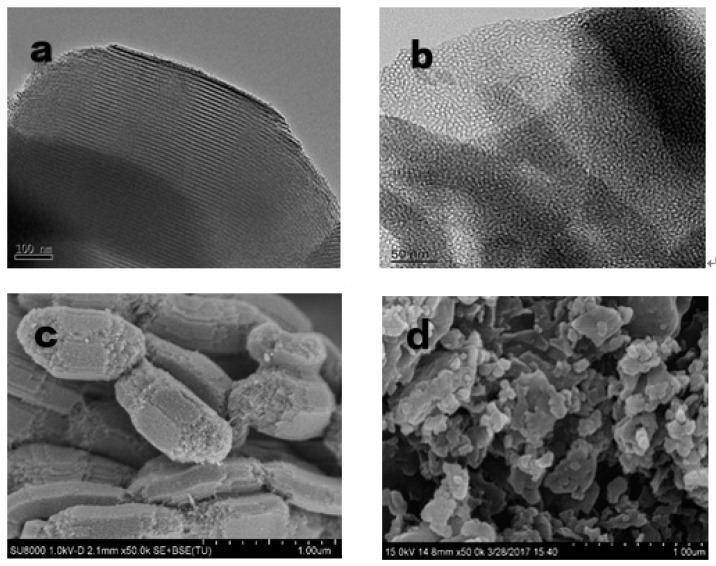
The TEM and SEM micrographs of SBA15 (**a**,**c**) and MCM-41 (**b**,**d**).

**Figure 2 materials-12-00546-f002:**
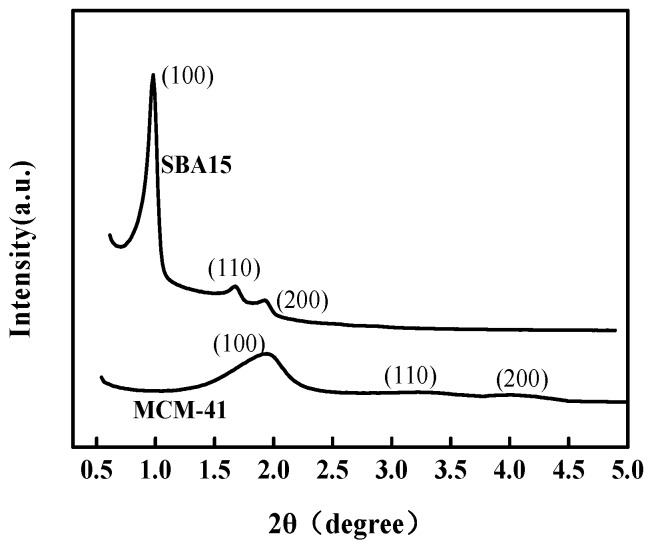
The small-angle X-ray diffraction patterns of SBA15 and MCM-41.

**Figure 3 materials-12-00546-f003:**
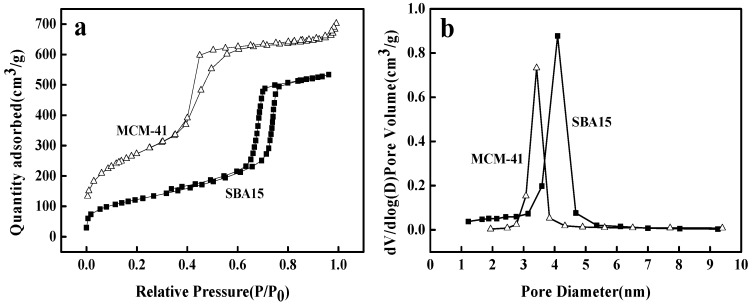
N2 adsorption/desorption isotherms (**a**) and corresponding BJH pore size distribution (**b**) of SBA15 and MCM-41.

**Figure 4 materials-12-00546-f004:**
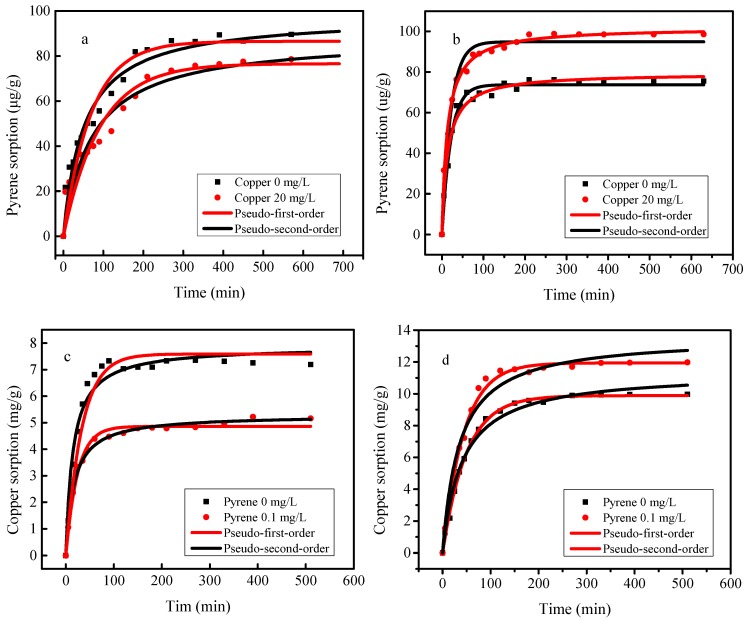
Adsorption kinetics of pyrene and copper onto SBA15 (**a**,**c**) and MCM-41 (**b**,**d**). (**a**,**b**) Pyrene adsorption in the single and binary system, experimental conditions: [pyrene]_0_ = 0.1 mg L^−1^, [copper]_0_ = 0 or 20 mg L^−1^, dosage = 0.5 g L^−1^; (**c**,**d**) Copper adsorption in the in the single and binary system, experimental conditions: [copper]_0_ = 20 mg L^−1^, [pyrene]_0_ = 0 or 0.1 mg L^−1^, dosage = 0.5 g L^−1^,.

**Figure 5 materials-12-00546-f005:**
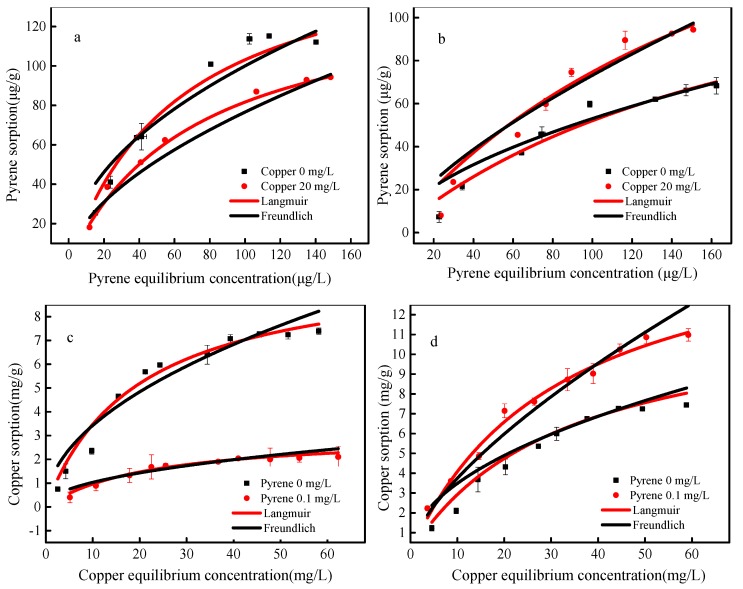
Adsorption isotherms of pyrene and copper onto SBA15 (**a**,**c**) and MCM-41 (**b**,**d**). (**a**,**b**) Pyrene adsorption in the single and binary system, experimental conditions: [pyrene]_0_ = 0.1 mg L^−1^, [copper]_0_ = 0 or 20 mg L^−1^, dosage = 0.5 g L^−1^; (**c**,**d**) Copper adsorption in the single and binary system, experimental conditions: [Copper]_0_ = 20 mg L^−1^, [pyrene]_0_ = 0 or 0.1 mg L^−1^, dosage = 0.5 g L^−1^.

**Figure 6 materials-12-00546-f006:**
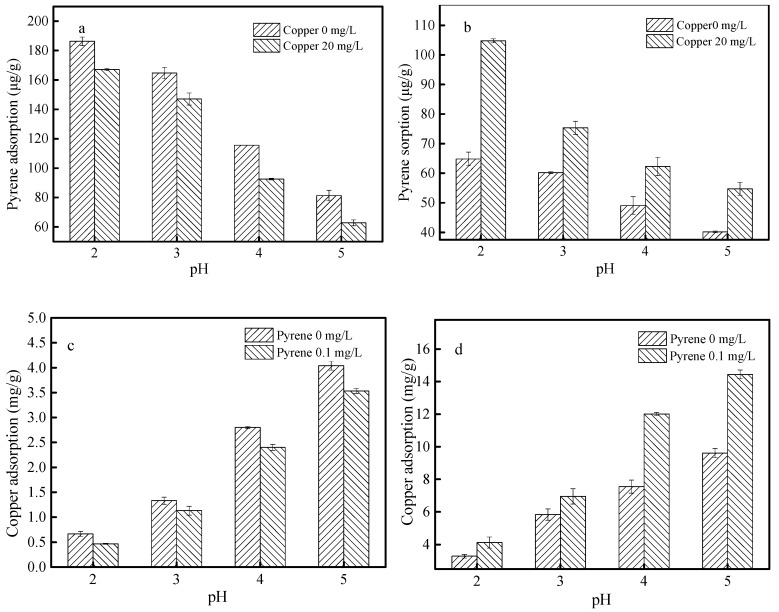
Effect of pH on the adsorption of pyrene and copper onto SBA15 (**a**,**c**) and MCM-41 (**b**,**d**). (**a**,**b**) Effect of pH on the pyrene adsorption in the single and binary system, experimental conditions: [pyrene]_0_ = 0.1 mg L^−1^, [copper]_0_ = 0 or 20 mg L^−1^, dosage = 0.5 g L^−1^; (**c**,**d**) Effect of pH on the copper adsorption in the single and binary system, experimental conditions: [copper]_0_ = 20 mg L^−1^, [pyrene]_0_ = 0 or 0.1 mg L^−1^, dosage = 0.5 g L^−1^.

**Figure 7 materials-12-00546-f007:**
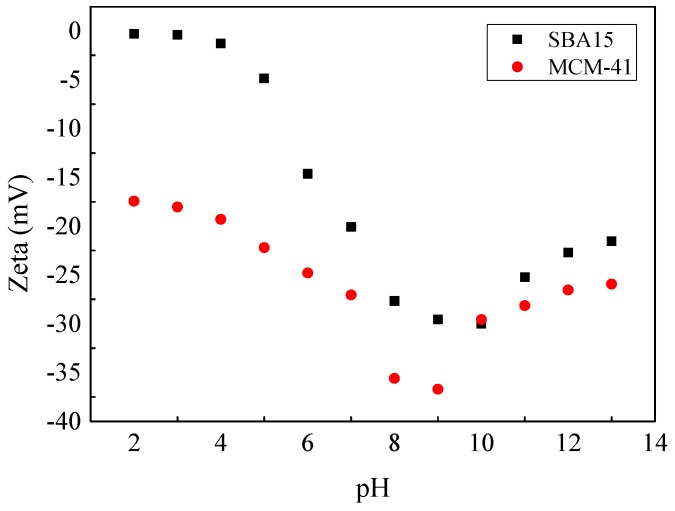
The zeta potential of SBA15 and MCM-41.

**Figure 8 materials-12-00546-f008:**
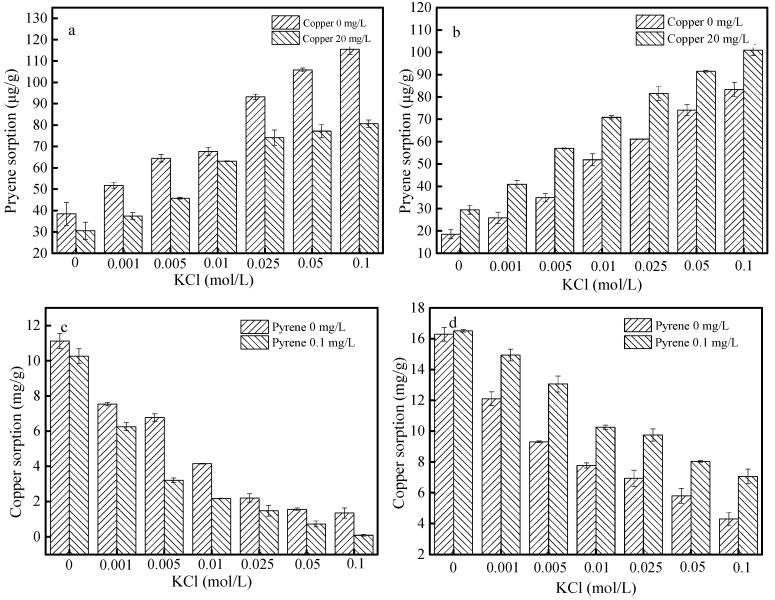
Effect of ionic strength on the adsorption of pyrene and copper onto SBA15 (**a**,**c**) and MCM-41(**b**,**d**). (**a**,**b**) Effect of ionic strength on the pyrene adsorption in the single and binary system, experimental conditions: [pyrene]_0_ = 0.1 mg L^−1^, [copper]_0_ = 0 or 20 mg L^−1^, dosage = 0.5 g L^−1^; (**c**,**d**) Effect of ionic strength on the copper adsorption in the single and binary system, experimental conditions: [copper]_0_ = 20 mg L^−1^, [pyrene]_0_ = 0 or 0.1 mgL^−1^, dosage = 0.5 g L^−1^.

**Figure 9 materials-12-00546-f009:**
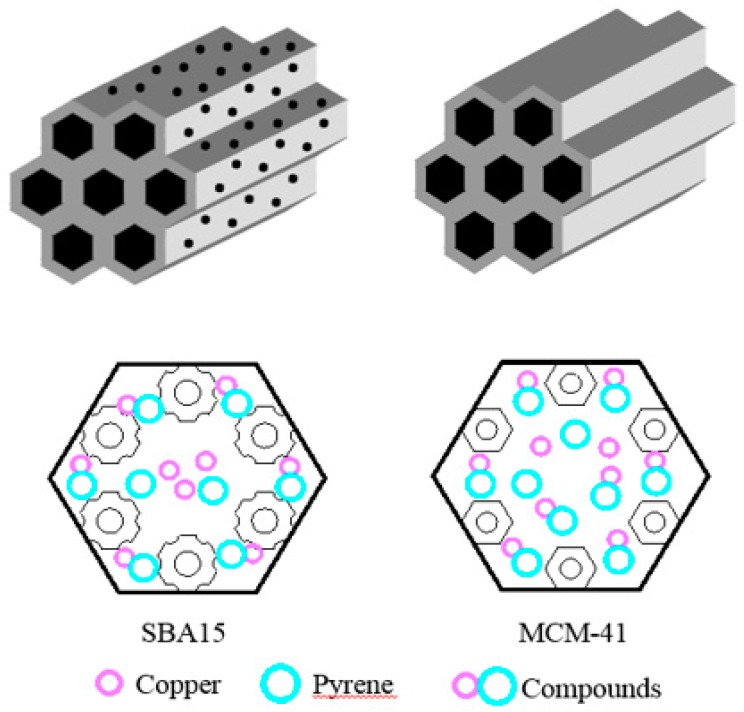
Three-dimensional model and adsorption schematic diagram of SBA15 and MCM-41 on pollutants.

**Table 1 materials-12-00546-t001:** Pore structure properties of SBA15 and MCM-41.

Materials	BET (m^2^ g^−1^)	Pore Volume (cm^3^ g^−1^)	Pore Size (nm)
SBA15	409.45	0.83	4.12
MCM-41	969.94	1.09	3.74

**Table 2 materials-12-00546-t002:** The kinetics parameters and correlation coefficients for the adsorption of pyrene and copper onto SBA15 and MCM-41.

Adsorbent	Initial Concentration (mg L^−1^)		Pseudo-First-Order			Pseudo-Second-Order		
			*Q_e_* (mg g^−1^)	*K* _1_	*R* ^2^	*Q_e_* (mg g^−1^)	*K*_2_ (10^−4^)	*R* ^2^
SBA15	Pyrene = 0.1	Copper = 0	0.087	0.014	0.94	0.098	1.96	0.96
		Copper = 20	0.077	0.011	0.92	0.089	1.56	0.95
	Copper = 20	Pyrene = 0	7.36	0.050	0.97	7.88	84.6	0.97
		Pyrene = 0.1	4.86	0.041	0.98	5.29	114	0.99
MCM-41	Pyrene = 0.1	Copper = 0	0.072	0.050	0.98	0.081	8.22	0.98
		Copper = 20	0.091	0.053	0.97	0.102	7.38	0.99
	Copper = 20	Pyrene = 0	9.22	0.028	0.97	11.80	19.1	0.99
		Pyrene = 0.1	11.31	0.030	0.97	14.10	18.4	0.97

**Table 3 materials-12-00546-t003:** The Langmuir and Freundlich parameters for the adsorption of pyrene and copper onto SBA15 and MCM-41.

Adsorbent	Initial Concentration (mg L^−1^)		Langmuir			Freundlich		
			*Q*_max_ (mg g^−1^*)*	*b* (L mg^−1^)	*R* ^2^	*n*	*K_f_*	*R* ^2^
SBA15	Pyrene = 0.1	Copper = 0	0.168	0.016	0.98	2.08	19.93	0.94
		Copper = 20	0.141	0.014	0.99	1.78	5.74	0.98
	Copper = 20	Pyrene = 0	10.28	0.051	0.98	2.01	1.09	0.94
		Pyrene = 0.1	3.080	0.045	0.96	0.35	2.12	0.90
MCM-41	Pyrene = 0.1	Copper = 0	0.157	0.005	0.97	1.77	3.91	0.94
		Copper = 20	0.198	0.005	0.97	1.44	2.96	0.97
	Copper = 20	Pyrene = 0	10.53	0.030	0.92	2.04	1.13	0.85
		Pyrene = 0.1	15.09	0.031	0.98	1.48	0.79	0.97

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
