# Peer review of "Systematic Investigation of the Synergistic and Antagonistic Effects on the Removal of Pyrene and Copper onto Mesoporous Silica from Aqueous Solutions"

_materials, 2019, doi:10.3390/ma12030546_

Round 1
Reviewer 1 Report
GENERAL COMMENT
Authors conducted the batch experiments aiming to explain the adsorption of pyrene and copper from aqueous solution by two different mesoporous materials based on silica (SBA15 and MCM-41). Adsorbents were prepared before experiment and their properties were characterized in details. The adsorption process was analyzed for individual chemicals (pyrene or Cu) and for their binary mixture in the some range of concentrations. Such approach focused on the chemicals mixture is scientifically interesting but the study is not of a high novelty. Very similar research was published by Zhang et al. (2017) [Zhang Z., Hou X., Zhang X., Li H. 2017. The synergistic adsorption of pyrene and copper onto Fe(III) functionalized mesoporous silica from aqueous solution. Colloids and Surfaces A: Physicochemical and Engineering Aspects, 520, 39-45.]. The way of preparing the work also raises many objections. The manuscript is poorly written, there are many unfinished and unclear sentences. Authors use many mental shortcuts, which make the manuscript difficult to understand. The article requires also language corrections. The work is very similar to the article published by Zhang et al. (2017), thus the paper needs significant re-working and I am recommending rejection.
SPECIFIC COMMENTS
Introduction
Line 31: please correct “meals” to “metals”
Line 32: the sentence is not clear “With different number of benzene rings, researcher confirmed that PAHs are toxic to livings directly and caused different cancer in variety of organs”. Depending on the number of benzene rings PAHs have different properties, such as water solubility, sorption affinity, volatility and toxicity. It will be useful for the reader to characterize here not the group of PAHs but rather pyrene – used in the experiment.
Lines 34-35: the following sentence is incomplete and unclear: “As a persistent material, heavy metals could not be microbial decomposition and cause significance pathological changes to organisms”
Lines 35-36: please correct “..could enter into water environment by various methods” to “could enter the aquatic environment from various sources”
Lines 36-37: please explain the statement: “Results showed that PAHs and heavy metals could be detected in various aqueous solution”. What kinds of aqueous solutions do you mean?
Lines 39-41: please reword the following sentence, which is not clear: “Furthermore, results showed that the PAHs and heavy metals could form complex compounds by cation-π bonging, which enhance the co-toxicity of PAHs and heavy metals than their individual toxicity”
Line 51: it should be: “carbon nanoporous materials”
Lines 63: it should be: Hu et al. [17] found that…
Line 64: it should be: Batati et al. [18] used…
Line 66: it should be: Mureseanu et al. [19] used…
Materials and methods
-Please provide the method used for the potential zeta determination (results presented in Figure 7).
-References for the BET and BJH methods should be included in the methodological section.
-Description of the adsorption experiments given in this section is incomplete and unclear, thus it is difficult to understand the results presented in the Figures. There is a lack of information: what the concentration of pyrene stock solution was? How many concentrations of pyrene and copper were tested? In which time intervals the measurements were performed?
Results and discussion
Authors findings are rather properly described, I suggest to extent the discussion of results.
Lines 198-199: the incomplete sentence: “The equilibrium time of pyrene and copper onto SBA15 was about.”
Line 244: it should be “..in both sole and binary systems..”
Lines 276-277: the following sentence is incomplete: “As a negative charged matter, pyrene showed negative charged mostly in the pH values.” Please rephrase and correct.
Lines 278-279: the following sentence is incomplete: “However, the copper showed positive charged mostly in the pH values.” Please correct and specify the values of pH.
Line 320: it should be Kosuge et al. [30]…
Lines 333-334: the sentence “The adsorption schematic diagram of copper and pyrene onto SBA15 and MCM-41 were depicted in the two adsorbents on pollutants are shown” is not clear – please correct and reword.
Author Response
Response to reviewer 1 Comments
Dear Reviewer,
We are very grateful to you for your professional comments on the original version. We appreciate the efforts made by you, which are greatly helpful to enhance the quality of our paper. According to your professional comments, we carefully revised the manuscript. The changes in the revision have been clearly marked in RED in PDF file. The point-by-point replies to your questions are listed as follows.
Reviewer #1:
Point 1: Line 31: please correct “meals” to “metals”
Response 1: Thank you for your criticism.
According to your suggestion, the word “meals” in page 1, line 31 was corrected to “metals”.
Point 2: Line 32: the sentence is not clear “With different number of benzene rings, researcher confirmed that PAHs are toxic to livings directly and caused different cancer in variety of organs”. Depending on the number of benzene rings PAHs have different properties, such as water solubility, sorption affinity, volatility and toxicity. It will be useful for the reader to characterize here not the group of PAHs but rather pyrene – used in the experiment
Response 2: Thank you very much for your suggestions, which is very helpful to improve the manuscript.
According to your suggestions, the sentences in page 1, line 32, were replaced by “As one of the typical PAHs, pyrene was selected as a priority pollutant by the US Environmental Protection Agency. Furthermore, pyrene has been widely used as an indicator compounds for the overall burden of all PAHs.”
Point 3: Lines 34-35: the following sentence is incomplete and unclear: “As a persistent material, heavy metals could not be microbial decomposition and cause significance pathological changes to organisms”
Response 3: Thank you for your professional advices, which is helpful to improve our manuscript.
According to your advices, in page 1, line 34-35, the sentences were replaced by “As a persistent material, heavy metals could not be eliminated by microbial decomposition. The heavy metals could be enriched by the food chain and may cause significance pathological changes to the organisms.”
Point 4: Lines 35-36: please correct “.could enter into water environment by various methods” to “could enter the aquatic environment from various sources”
Response 4: Thank you very much for your advices.
According to your advices, the sentences now at Page 1, Line 37 was corrected as: “..could enter the aquatic environment from various sources”.
Point 5: Lines 36-37: please explain the statement: “Results showed that PAHs and heavy metals could be detected in various aqueous solution”. What kinds of aqueous solutions do you mean?
Response 5: Thank you very much for your questions.
According to your questions, the sentences at Page 1, Line 37 was changed as follows:
“Results showed that PAHs and heavy metals could be detected in various aquatic environment such as river, lakes and oceans.”
Point 6: Lines 39-41: please reword the following sentence, which is not clear: “Furthermore, results showed that the PAHs and heavy metals could form complex compounds by cation-π bonging, which enhance the co-toxicity of PAHs and heavy metals than their individual toxicity”
Response 6: Thank you very much for your suggestions.
According to your suggestions, the sentences at Page 1, Line 40-41 was replaced by: “Furthermore, results showed that the PAHs and heavy metals could form complex compounds by cation-π bonging, the toxicity of formed complex was larger than their individual toxicity which enhance the co-toxicity of PAHs and heavy metals.”
Point 7: Line 51: it should be: “carbon nanoporous materials”
Response 7: Thank you very much for your suggestions.
According to your suggestions, the words now at Page 2, Line 55 were corrected as “carbon nanoporous materials”.
Point 8: Lines 63: it should be: Hu et al. [17] found that…
Response 8: Thank you very much for your suggestions.
According to your suggestions, the sentences now at Page 2, Line 66 were corrected as: “Hu et al. [18] found that MCM-41 had a great adsorption capacity of phenanthrene and could be used for the removal of phenanthrene from aqueous solution”.
Point 9: Line 64: it should be: Batati et al. [18] used…
Response 9: Thank you very much for your suggestions.
According to your suggestions, the sentences now at Page 2, Line 68 were corrected as follows: “Batati et al. [10] used NH2-SBA15 organic-inorganic nanohybrid materials to eliminated PAHs from wastewater and found that these materials also show high adsorption capacity after several cycles of adsorption-desorption”.
Point 10: Line 66: it should be: Mureseanu et al. [19] used…
Response 10: Thank you very much for your suggestions.
According to your suggestion, the sentences now at Page 2, Line 70 were corrected as follows: “Mureseanu et al. [19] used N-propylsalicylaldimino-functionalized SBA15 to eliminate heavy metals and found the adsorbent has remarkable adsorption capacities in the removal of copper, cobalt, nickel and zinc”.
Point 11: Line 320: it should be Kosuge et al. [30]…
Response 11: Thank you very much for your suggestions.
According to your suggestion, the sentences now at Page 12, Line 323 were corrected as: “Kosuge et al. [30] found that in the adsorption of VOCs onto mesoporous silica, the adsorption process was prefer to occurred through micropores on SBA15 while through mesoporous on MCM-41”.
Point 12: Please provide the method used for the potential zeta determination (results presented in Figure 7).
Response 12: Thanks for your thoughtful suggestions.
According to your suggestions, the Zeta potential method were added in Page 4, Line 149-152, as follows: “The amount of 0.025 g prepared adsorbents (SBA15 and MCM41) were added into a 25 mL solution. The solution pH was kept at desired values in the range of 2.0 - 13.0 using HCl and NaOH. The mixtures were shaken for 24 h to allow it to reach the equilibrium and the Zeta potential was measured using a Nano Particle Sizing & Zeta potential Analyzer (Beckman Coulter, U.S.A)”.
Point 13: References for the BET and BJH methods should be included in the methodological section.
Response 13: Thank you for your professional comments.
According to your comments, the reference for the BET and BJH methods were added as [22] and [23]. The sentences at Page 3, Line 113 were changed as follows: “The specific surface area was calculated by Brunauer-Emmett-Teller (BET) method [22] pore volume and pore size distribution were calculated by BJH method [23].”
Point 14: Description of the adsorption experiments given in this section is incomplete and unclear, thus it is difficult to understand the results presented in the Figures. There is a lack of information: what the concentration of pyrene stock solution was? How many concentrations of pyrene and copper were tested? In which time intervals the measurements were performed?
Response 14: Thank you for your professional questions.
1) According to your comments, the concentration of pyrene stock solution was added at Page 4, Line 135. The sentences were added as follows: “The 10 mg L-1 pyrene (methanol as a solvent) were prepared.”
2) In the adsorption isotherm experiments of pyrene and copper on SBA15 and MCM-41, the initial concentrations of pyrene are 0.025, 0.050, 0.075, 0.100, 0.125, 0.150 mg L-1. The concentration of copper intended to be designed are 5, 10, 15, 20, 25, 30, 35, 40, 50 mg L-1. When the solution is configured, the actually concentration of the contaminant may be biased to the theoretical value. The concentration of contaminants in our manuscript were based on actual measurements.
In addition, according to your questions, the sentences at Page 4, Line 136-138 were changed as follows: “Then the stock solution of pyrene were diluted to 0.025, 0.050, 0.075, 0.100, 0.125, 0.150 mg L-1 which contains 0 or 20 mg L−1 copper. For the adsorption of copper, the solution of copper with the concentration of 5, 10, 15, 20, 25, 30, 35, 40, 50 mg L-1 which contains 0 or 0.1 mg L−1 pyrene.”
3) In the kinetics experiments, the time intervals for measurements are 0, 5, 10, 15, 25, 35, 45, 60, 75, 90, 120, 150, 180, 210, 270, 330, 390, 510, 630 min.
Besides according to your questions, the sentences at Page 3, Line 126-128 were changed as follows: “At 0, 5, 10, 15, 25, 35, 45, 60, 75, 90, 120, 150, 180, 210, 270, 330, 390, 510, 630 min, the samples were taken and centrifuged for 2 min at 5000 rpm.”
Point 15: Lines 198-199: the incomplete sentence: “The equilibrium time of pyrene and copper onto SBA15 was about.”
Response 15: Thank you for your advices. According to your advices, “The equilibrium time of pyrene and copper onto SBA15 was about.” now at Page 7, Line 206 were corrected as: “The equilibrium time of pyrene and copper onto SBA15 was about 200 and 350 min.”
Point 16: Line 244: it should be “..in both sole and binary systems..”
Response 16: Thank you for your advices. According to your advices, the sentence now at Page 9, Line 250 were corrected as: “in both sloe and binary systems.”
Point 17: Lines 276-277: the following sentence is incomplete: “As a negative charged matter, pyrene showed negative charged mostly in the pH values.” Please rephrase and correct.
Response 17: Thank you for your professional comments. According to your comments, the sentence now at Page 11, Line 281 were corrected as follows: “As a nonionic compound, pyrene has electron-rich π ring (as a donor of electrons).”
Point 18: Lines 278-279: the following sentence is incomplete: “However, the copper showed positive charged mostly in the pH values.” Please correct and specify the values of pH.
Response 18: Thanks for your thoughtful suggestions. According to your suggestions, the sentence now at Page 11, Line 283 were corrected as follows: “However, the copper showed positive charged in the pH values of 2.0 to 5.0.”
Point 19: Lines 333-334: the sentence “The adsorption schematic diagram of copper and pyrene onto SBA15 and MCM-41 were depicted in the two adsorbents on pollutants are shown” is not clear – please correct and reword.
Response 19: Thank you for your professional comments. According to your comments, the sentence now at Page 13, Line 336 were corrected as follows: “The adsorption schematic diagram of copper and pyrene onto SBA15 and MCM-41 were showed in Fig. 9.”
Other changes in the revision have been marked in RED in PDF file.
Thank you very much again for your comments on this manuscript.
Reviewer 2 Report
-The authors should comment on the practical applicability of the proposed adsorbents in the wastewater treatments.
-The possible matrix effect in real samples should be explained. The limitations and drawbacks of the proposed methodology should be critically assessed in the manuscript.
-The rationale for the selection of the concentration range should be explicitly mentioned in the manuscript. It should be demonstrated that the concentration range is practically relevant (why 0.1 mg/L for pyrene and 20 mg/L for copper?).
-The state of the art should be improved in the introduction section, specifically in the section relative to the adsorbents used for the removal of the contaminants. For example, I suggest to add the following recent papers:
Petrella, A., Petruzzelli, V., Ranieri, E., Catalucci, V., Petruzzelli, D. (2016). Sorption of Pb (II), Cd (II), and Ni (II) from single-and multimetal solutions by recycled waste porous glass. Chemical Engineering Communications, 203(7), 940-947.
Petrella, A., Spasiano, D., Acquafredda, P., De Vietro, N., Ranieri, E., Cosma, P., Rizzi, V, Petruzzelli, V., Petruzzelli, D. (2018). Heavy metals retention (Pb (II), Cd (II), Ni (II)) from single and multimetal solutions by natural biosorbents from the olive oil milling operations. Process Safety and Environmental Protection, 114, 79-90.
-A deeper study on the Cu/pyrene complexation must be reported. How it can be chemically explained? Which experimental technique can show this?
-Error bars are reported for most of the experiments. Their derivation should be clearly stated in the experimental section. How many experiments were performed? How did the authors calculate the errors? Error bars are not reported in Fig. 4. Clarify in the text.
-I suggest to improve the conclusion section with all the relevant results obtained.
For this reason, major revisions are required
Author Response
Response to reviewer 2 Comments
Dear Reviewer,
We are very grateful to you for your professional comments on the original version. We appreciate the efforts made by you, which are greatly helpful to enhance the quality of our paper. According to your professional comments, we carefully revised the manuscript. The changes in the revision have been clearly marked in RED in PDF file. The point-by-point replies to your questions are listed as follows.
Reviewer #2:
Point 1: The authors should comment on the practical applicability of the proposed adsorbents in the wastewater treatments.
Response 1: Thank you for your professional comments.
As one of the typical adsorbent, mesoporous silica has great potential application for the elimination of pollutants from aqueous solutions. Due to the large surface area, tunable pore structure and well-understood surface chemistry, the mesoporous silica could offer lots of useful adsorption sites for the removal of many pollutants, such as pharmaceuticals, phosphate, and heavy metals. In this study, to investigate the synergistic and antagonistic effects on the removal of pyrene and copper, two typical mesoporous adsorbents were selected. In our opinion, the first step was investigated the synergistic and antagonistic effects of pyrene and copper in batch experiments. The next step was to study the practical applicability of the proposed adsorbents in the wastewater treatments. About the practical applicability of mesoporous silica in the wastewater treatments, researchers have used typical mesoporous silica for the removal of many pollutants from wastewater such as Cu, Ni, Cr and Hg [1, 2]. Therefore, we believe that mesoporous materials could be effectively applied in wastewater treatments.
Point 2: The possible matrix effect in real samples should be explained. The limitations and drawbacks of the proposed methodology should be critically assessed in the manuscript.
Response 2: Thanks for your professional suggestions. The particle effect may be existed in real samples. In the manuscript, SBA15 and MCM-41 with different structure and pore size have different adsorption effects on copper and pyrene in the binary system. The limitations and drawbacks of the proposed methodology were assessed in the manuscript at Page 2, Line 49-51 “Extraction technique and advanced oxidation process usually need high investment and high maintenance costs, while biodegradation and photocatalysis oxidation processes may release some secondary by-products which are carcinogenic and mutagenic compounds.”
Point 3: The rationale for the selection of the concentration range should be explicitly mentioned in the manuscript. It should be demonstrated that the concentration range is practically relevant (why 0.1 mg/L for pyrene and 20 mg/L for copper?).
Response 3: Thanks for your thoughtful suggestions.
In order to study the adsorption process of pyrene and copper onto two adsorbents for, the selected concentrations of pyrene and copper were higher than the actual concentration in environment. In fact, the concentration of heavy metal was higher than pyrene concentration in actual aqueous soltions. Hence, in this manuscript the concentration of copper and pyrene were selected as 20 mg/L and 0.1 mg/L.
Point 4: The state of the art should be improved in the introduction section, specifically in the section relative to the adsorbents used for the removal of the contaminants.
Response 4: Thanks for your professional suggestions. According to you suggestions, we cited some new references in the introduction at Page2, Line 55, “carbon nanoporous materials [9]”, “porous galss [10]”, “biosorbent [6]”.
Point 5: A deeper study on the Cu/pyrene complexation must be reported. How it can be chemically explained? Which experimental technique can show this?
Response 5: Thank you for your professional comments. To further investigate the interaction between Cu (II) and pyrene, the Cu (II) ion-selective electrode was used to analyze the amount of free Cu (II) ion in solution without adsorbents. The results were showed as below. The results showed that the presence of pyene decreased the amount of free Cu (II) ion in the aqueous solution, which suggested that the complexes of pyrene-Cu (II) may formed.
Free Cu (II) concentration with or without pyrene
| Pyrene concentrations (mg L-1) | Cu (II) concentrations (mg L-1) | |||
| 0 | 0.90 | 1.52 | 3.64 | 6.33 |
| 0.1 | 0.62 | 1.28 | 3.16 | 5.71 |
Point 6: Error bars are reported for most of the experiments. Their derivation should be clearly stated in the experimental section. How many experiments were performed? How did the authors calculate the errors? Error bars are not reported in Fig. 4. Clarify in the text.
Response 6: Thanks for your professional suggestions.
1) The standard deviation in the experiments were state at Page 3, Line 118-122, “All the experiments were performed in duplicate and the values listed were calculated by the average and standard deviation. The standard deviation were calculated by Eq. (1).
(1)
Where S is the standard deviation, `x is the average of each sample, xi is the ith sample.”
2) All the experiments were performed in duplicate except the kinetic experiments.
3) We are very sorry that we did not take parallel samples in the kinetic experiments. So Error bars are not reported in Fig. 4. However, the adsorption kinetic were used to investigated the adsorption process of pyrene and copper onto mesoporous. The results of adsorption kinetic showed the tendency of adsorption. In the manuscript, all the adsorption kinetics showed the same models for in both solo and binary systems. Furthermore, other researcher showed the same results of adsorption pollutants in mesoporous silica. Therefore we believe the data showed in the adsorption kinetics were credible.
Point 7: I suggest to improve the conclusion section with all the relevant results obtained.
Response 7: Thanks for your professional suggestions. According to you suggestions, the conclusion section were replaced by “In this study, two typical mesoporous silica SBA15 and MCM-41were prepared and used to investigate the removal of pyrene and copper from aqueous solution simultaneously. Results showed that the adsorption of pyrene and copper onto mesoporous silica was fast in both sole and binary systems. The Langmuir model fitted the adsorption isotherms better than the Freundlich model which suggested that the adsorption was probably monolayer molecular adsorption. . The kinetics of pyrene and copper onto SBA-15 and MCM-41 could be satisfactorily described by pseudo-second-order model which suggested that the adsorption may be chemisorption. The solution pH values and ionic strength affect the adsorption process greatly. Furthermore, the synergistic effects could be observed on the adsorption of pyrene and copper onto MCM-41, while the antagonistic effects could be observed on the adsorption of pyrene and copper onto SBA15. The different pore sizes have a great influence on the adsorption process of pyrene, copper and their composite contaminants. This study could provide new ideas for the synergistic removal of PAHs and heavy metals in water environments”
References:
1. Mahsa Tahergorabi, A.E., Majid Kermani, Mehdi Shirzad-Siboni, Application of thiol-functionalized mesoporous silica-coated magnetite nanoparticles for the adsorption of heavy metals. Desalination and Water Treatment, 2015. 1: p. 1944-1957.
2. Pérez-Quintanilla, D., A. Sánchez, and I. Sierra, Preparation of hybrid organic-inorganic mesoporous silicas applied to mercury removal from aqueous media: Influence of the synthesis route on adsorption capacity and efficiency. Journal of Colloid and Interface Science, 2016. 472: p. 126-134.
Other changes in the revision have been marked in RED in PDF file.
Thank you very much again for your comments on this manuscript.

Round 2
Reviewer 1 Report
GENERAL COMMENT
Authors conducted the interesting research on the adsorption of pyrene and copper from aqueous solution by two different mesoporous materials. In the laboratory experiment their try to explain the mechanisms of contaminants removal as a sole substances or in the mixture.
I have had a lot of comments and recommendations previously, authors have addressed all of them. Many corrections have been made in the manuscript, however the authors did not avoid mistakes. Thus, I recommend to publish article after minor revision.
SPECIFIC COMMENTS
Line 48: please correct “approachs” to “approaches”
Line 53: it should be “adsorption is attractive”
Line 56: please correct “porous galss” to “porous glass”
Line 69: please correct “.. to eliminated PAHs from” to “..to eliminate PAHs from”
Line 80: replace “structural” to “structure”
Line 82: it should be “To investigate..”
Lines 88-92: please rephrase this paragraph. It was more clear in the previous version of the manuscript.
Line 94: please explain the abbreviation TEOS
Line 101: explain CTAB
Lines 119-120: the sentence “The experiments with a background solution in 0.01 mol L−1 KCl and carried out at 25 °C” is not clear
Line 163: it should be “are shown”
Line 166: correct “domains with are uniform length of 1 μm” to “domains with a uniform length of 1 μm”
Line182: change “refection intensity” to “reflection intensity”
Line 194: correct “These datas” to “These data”
Lines 212-214: the following sentence is not clear, please rephrase “To research the adsorption behavior further, using the pseudo-first-order and pseudo-second order models to analyze the experimental data, the equations of them represented by Eqs. (3) and (4)”
Line 247: correct unit (mg g-1)
Line 251: it should be “in both sole and binary systems”
Line 325: it should be “the adsorption process was prefer to occur”
Author Response
Response to reviewer 1 Comments
Dear Reviewer,
We are very grateful to you for your professional comments on the original version. We appreciate the efforts made by you, which are greatly helpful to enhance the quality of our paper. According to your professional comments, we carefully revised the manuscript. The changes in the revision have been clearly marked in RED in PDF file. The point-by-point replies to your questions are listed as follows.
Reviewer #1:
Point 1: Line 48: please correct “approachs” to “approaches”
Response 1: Thank you for your criticism.
According to your suggestion, the word “approachs” in page 2, line 48 was corrected to “approaches”.
Point 2: Line 53: it should be “adsorption is attractive”
Response 2: Thank you very much for your suggestions, which is very helpful to improve the manuscript.
According to your suggestions, the sentences in page 2, line 53, were replaced by “Among these methods, adsorption is attractive due to the multiple benefit of high removal capacity, easy operation, energy savings, quickly and widely application.”
Point 3: Line 56: please correct “porous galss” to “porous glass”
Response 3: Thank you for your professional advices, which is helpful to improve our manuscript.
According to your advices, the words “porous galss” in page 2, line 56 were corrected to “porous glass”
Point 4: Line 69: please correct “.. to eliminated PAHs from” to “..to eliminate PAHs from”
Response 4: Thank you very much for your suggestions, which is very helpful to improve the manuscript.
According to your suggestions, the sentences “.. to eliminated PAHs from” in page 2, line 69, were corrected to “..to eliminate PAHs from”
Point 5: Line 80: replace “structural” to “structure”
Response 5: Thank you for your criticism.
According to your suggestion, the word “structural” in page 2, line 80 was corrected to “structure”.
Point 6: Line 82: it should be “To investigate..”
Response 6: Thank you very much for your suggestions, which is very helpful to improve the manuscript.
According to your suggestions, the sentences in page 2, line 82, were replaced by “To investigate the adsorption behavior and effects such as kinetics, isotherms, ionic strength and pH effect the batch experiments were conducted in sole and binary systems.”
Point 7: Lines 88-92: please rephrase this paragraph. It was more clear in the previous version of the manuscript.
Response 7: Thank you very much for your suggestions.
According to your suggestions, this paragraph at Page 2, Line 88-92 were replaced by: “Pyrene (99%) was purchased from the Sigma Aldrich Chemical Co (WI, USA). Tetraethyl silicate (TEOS, 98%), PluronicP123 (PEO20 PPO70 PEO20, Mav = 5800) and cetyl trimethyl ammonium bromide (CTAB, 99%) were purchased from Alfa Aesar Corp. UPLC grade of Acetonitrile and methanol were purchased from Fisher Scientific Corp. Other chemicals including HCl, NaOH, CuCl2·2H2O, KCl and ethanol were all of analytical grade. All solutions were prepared in high-purity water (Milli-Q).”
Point 8: Line 94: please explain the abbreviation TEOS
Response 8: Thank you for your professional comments.
TEOS is the abbreviation of Tetraethyl silicate. According to your comments, the sentences now in page 3, line 95, were replaced by “SBA15 was synthesized by hydrothermal method with Tetraethyl silicate (TEOS) as the silicon source and P123 as template.”
Point 9: Line 101: explain CTAB
Response 9: Thank you for your professional comments.
CTAB is the abbreviation of cetyl trimethyl ammonium bromide. According to your comments, the sentences now in page 3, line 102, were replaced by “MCM-41 was prepared by a direct-synthesis approach under alkaline condition using Tetraethyl silicate (TEOS) as the silicon source and cetyl trimethyl ammonium bromide (CTAB) as the template.”
Point 10: Lines 119-120: the sentence “The experiments with a background solution in 0.01 mol L−1 KCl and carried out at 25 °C” is not clear
Response 10: Thank you for your professional comments.
According to your comments, the sentences now at Page 3, Line 122 were corrected as:” The experiments were carried out at 25 °C with an ionic strength of 0.01 mol L−1 KCl.”
Point 11: Line 163: it should be “are shown”
Response 11: Thank you very much for your suggestions, which is very helpful to improve the manuscript.
According to your suggestions, the words now in page 5, line 166, were corrected as: “are shown”
Point 12: Line 166: correct “domains with are uniform length of 1 μm” to “domains with a uniform length of 1 μm”
Response 12: Thank you very much for your suggestions, which is very helpful to improve the manuscript.
According to your suggestions, the sentences now in page 5, line 169, were corrected as: “domains with a uniform length of 1 μm”
Point 13: Line182: change “refection intensity” to “reflection intensity”
Response 13: Thank you very much for your suggestions, which is very helpful to improve the manuscript.
According to your suggestions, the words “refection intensity” now in page 5, line 185, were corrected as: “reflection intensity”.
Point 14: Line 194: correct “These datas” to “These data”
Response 14: Thank you very much for your suggestions, which is very helpful to improve the manuscript.
According to your suggestions, the words “These datas” now in page 6, line 197, were corrected as: “These data”.
Point 15: Lines 212-214: the following sentence is not clear, please rephrase “To research the adsorption behavior further, using the pseudo-first-order and pseudo-second order models to analyze the experimental data, the equations of them represented by Eqs. (3) and (4)”
Response 15: Thank you for your professional comments.
According to your comments, the sentences now at Page 7, Line 215-217 were replaced by: “In order to study the adsorption kinetics further, pseudo-first-order and pseudo-second-order models were used to predict the adsorption mechanism, the equations were expressed by Eqs. (3) and (4), respectively.”
Point 16: Line 247: correct unit (mg g-1)
Response 16: Thank you very much for your suggestions, which is very helpful to improve the manuscript.
According to your suggestions, the words now in page 9, line 250, were corrected as: “mg g-1”.
Point 17: Line 251: it should be “in both sole and binary systems”
Response 17: Thank you for your professional comments.
According to your comments, the sentences now at Page 9, Line 254 were corrected as: “in both sole and binary systems”
Point 18: Line 325: it should be “the adsorption process was prefer to occur”
Response 18: Thank you for your professional comments.
According to your comments, the sentences now at Page 12, Line 328 were corrected as: “the adsorption process was prefer to occur”
Other changes in the revision have been marked in RED in PDF file.
Thank you very much again for your comments on this manuscript.

Reviewer 2 Report
The paper can be now accepted for the publication
Author Response
Thank you very much for your comments on this manuscript entitled “Systematic investigation of the synergistic and antagonistic effects on the removal of pyrene and copper onto mesoporous silica from aqueous solutions”. According to the comments from you, we carefully checked and completed the necessary corrections in the submitted revised manuscript. We believe that our manuscript has improved by addressing to your comments.
Thank you very much again for your comments on this manuscript.